# FaAlGrad: Fairness through Alignment of Gradients across Different Subpopulations

**Nikita Malik**                                                                          *nikitamalik2303@gmail.com*
*Department of Information Technology*
*Manipal Institute Of Technology, Manipal*

**Konda Reddy Mopuri**                                                                    *krmopuri@ai.iith.ac.in*
*Department of Artificial Intelligence*
*Indian Institute of Technology Hyderabad.*

**Reviewed on OpenReview:** *https://openreview.net/forum?id=k4AxEwTaHq*

## Abstract

The growing deployment of Machine Learning systems has increased interest in systems optimized for other important criteria along with the expected task performance. For instance, machine learning models often exhibit biases that lead to unfair outcomes for certain protected subpopulations. This work aims to handle the bias in machine learning models and enhance their fairness by aligning the loss gradients. Specifically, leveraging the meta-learning technique, we propose a novel training framework that aligns the gradients computed across different subpopulations for learning fair classifiers. Aligning the gradients enables our framework to regularize the training process, thereby prioritizing fairness over predictive accuracy. Our experiments on multiple benchmark datasets demonstrate significant improvements in fairness metrics without having any exclusive regularizers for fairness. Thus our work contributes to developing fairer machine learning models with broader societal benefits. Sample code for implementing the proposed framework is available at this link.

## 1 Introduction

Machine learning models have become integral to numerous real-world applications, from credit approval and hiring decisions to personalized recommendations and medical diagnoses. However, the widespread deployment of these models has raised concerns (e.g., Miller (2020); Stevenson (2018)) about their fairness and potential biases, particularly when used in high-stakes decision-making processes that impact individuals' lives and opportunities. Biases in machine learning models can stem from various sources, including biased training data, algorithmic design choices, and societal prejudices. Such biases can lead to discriminatory outcomes, disproportionately affecting certain demographic groups and reinforcing existing inequalities.

The need for fair and unbiased machine learning models has recently gained significant attention in academia and industry (e.g., Barocas et al. (2019); Chouldechova & Roth (2018); Zliobaite (2015); Romei & Ruggieri (2013)). Ensuring fairness in machine learning is a critical ethical concern and has become a growing area of research. Researchers and practitioners have explored various fairness-aware learning techniques to mitigate bias and promote equitable decision-making. However, achieving fairness often entails a trade-off with predictive accuracy, as many fairness-aware methods tend to compromise model performance. Various approaches have been proposed to address this issue, including pre-processing techniques (e.g., Sun et al. (2022); Madras et al. (2018)), in-processing fairness-aware algorithms (e.g., Biswas & Rajan (2021); Kamiran & Calders (2012)), and post-processing methods (e.g., Nandy et al. (2022); Kim et al. (2018)). However, these traditional methods often fail to comprehensively tackle the challenge of bias reduction, as they treat fairness as an isolated concern rather than a core aspect of the learning process.

In this paper, we identify a specific notion of bias in machine learning and present a framework to alleviate it. While instances of dataset bias and feature (variable) bias are prominent, we acknowledge the broader spectrum of biases that infiltrate machine learning models. Biases can manifest from various other factors including biased sampling, algorithmic design choice and evaluation biases.

- **Dataset bias**: if the dataset used for training the machine learning model does not represent the real-world population faithfully, the resulting model will likely be unfair to certain subpopulations.

- **Feature bias**: if the model considers certain features more important (e.g., for predictive accuracy), it may focus more on them to the detriment of other relevant features. Often feature bias is a consequence of dataset bias.

Therefore, one way to examine these biases is that they result from the dominance of specific subpopulations (or their representative feature attributes) during the learning process. We hypothesize that carefully alleviating that dominance can positively affect the fairness of the resulting machine learning systems. This paper proposes a novel gradient descent-driven training framework to align the loss gradients computed across different subpopulations in the training dataset. Aligning the gradients acts as a check on the bias-causing dominance of the responsible subpopulations. This, in turn, gives importance to samples of all the subpopulations. Often, the subpopulations are classified as *protected* or *unprotected* based on the values assumed by the sensitive features. This classification depends on various factors such as the task, distribution of the data samples and their targets, etc. It is important to note that the proposed method is particularly well suited for addressing feature and dataset biased in machine learning models.

The proposed approach achieves gradient alignment in the framework of meta-learning. Specifically, it is inspired by the Model-Agnostic Meta-Learning (MAML) Finn et al. (2017), a popular meta-learning algorithm that has demonstrated impressive performance across various applications. It enables models to acquire knowledge from multiple tasks and generalize effectively to new tasks. We leverage the flexibility of MAML and adapt it to address fairness concerns. Gradient alignment is achieved (refer to sec. 4.1) by choosing different subpopulations of the training data as separate tasks within the meta-learning framework, leading to the learning of fair models.

The objective of fair learning is to ensure similar model behavior toward both protected and unprotected subpopulations. The proposed approach suppresses the unwanted dominance caused by any subpopulations by aligning the loss gradients computed from both subpopulations. In other words, it avoids the potential interference of these gradients, resulting in a biased model. Specifically, the proposed meta-learning-inspired approach treats the protected and unprotected subpopulations as the meta-train and meta-test tasks, respectively (refer to sec. 3.1).

To evaluate the effectiveness of our approach, we conducted extensive experiments on four well-known classification datasets: the compas, Communities and Crimes, and the Adult dataset. These datasets are widely used to study fairness issues in machine learning. The experiments measure fairness metrics, such as demographic parity and equalized odds (refer to sec. 3.2), along with classification accuracy. The results of our experiments demonstrate that our meta-learning-inspired approach leads to significant improvements in fairness metrics, indicating a reduction in bias.

In summary, the key contributions of this paper are as follows:

- We propose a novel meta-learning inspired training strategy for mitigating bias in gradient descent-driven classifiers. It improves fairness by aligning the loss gradients computed across different subpopulations in the training dataset.

- We validate the effectiveness of the proposed approach in mitigating bias by evaluating on multiple real-world datasets.

- We demonstrate that the proposed method offers flexibility in achieving different levels of fairness and accuracy (refer to sec. 5.5). We can exercise a controlled trade-off between them by manipulating the proportions of the subpopulations in the MAML framework.

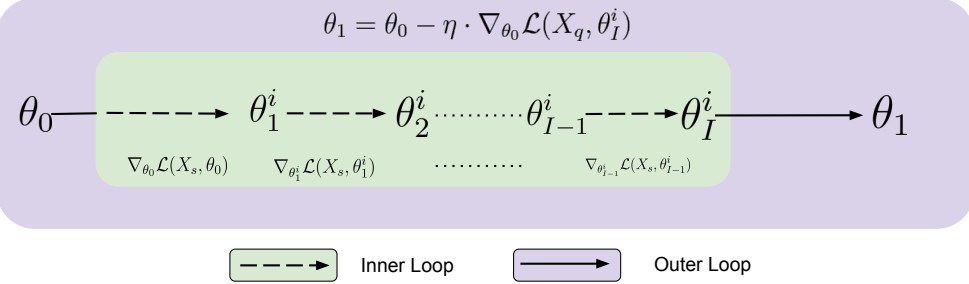

Figure 1: An overview of our meta-learning driven framework used to improve fairness. The green area denotes the inner loop or meta-train stage, while the purple area denotes the outer loop or the meta-test stage. Note that multiple inner loop iterations (I) are illustrated along with their update terms.

In the following sections, we review existing literature on fairness-aware learning methods. This will be followed by an introduction to MAML algorithm and various fairness metrics used in the paper. Subsequently, we will discuss the proposed meta-learning driven framework. Later, we present the experimental evaluation.

## 2 Related Works

To improve the fairness in machine learning models, several approaches have been designed inspired by the foundational work of Agarwal et al. (2018). These strategies involve Lagrangian relaxation and innovative validation or optimization techniques, collectively contributing to the advancements of fairness- aware machine learning models. This approach framed the challenge of achieving accurate classifiers under fairness constraints as a two-player zero-sum game. The underlying concept involves employing Lagrangian relaxation to handle the constrained optimization task. The first player, referred to as the $\theta$-player, fine-tunes the model's parameters by optimizing the objective function using the current Lagrange multipliers. Conversely, the second player, denoted as the $\lambda$-player, approximates the most robust Lagrangian relaxation through iterative updates to the Lagrange multipliers.

To mitigate the risk of overfitting fairness constraints, Cotter et al. (2019) devised a strategy in which the $\lambda$-player refines the Lagrange multipliers based on fairness violations quantified using a distinct validation set, rather than relying on the training set itself. Meanwhile, Mandal et al. (2020) proposed a unique approach where the $\lambda$-player employs linear programming techniques to compute the maximum potential fairness violation across various re-weightings of the training dataset. This method falls under Distributionally Robust Optimization (DRO) techniques. Maheshwari & Perrot (2023) uses constrained optimization and re-weighting techniques to enforce fairness. It minimizes error rates while adhering to fairness constraints using Lagrange multipliers and focuses on deriving a single optimal model, updating Lagrange multipliers via projected gradient descent and optimizing model parameters using a weighted sum of group-wise error rates.

After the introduction of MAML Finn et al. (2017), apart from numerous other applications, it has also been used in enhancing fairness. Along similar lines, the work of Slack et al. (2020) delves into the scenario where a fair model might display unfair behavior when deployed to separate places in the real world. This study introduces two notable contributions: Fairness Warnings and Fair-MAML. Fairness Warnings aim to predict if covariate shifts will lead to violating fairness. To achieve this, new training data is generated from the same distribution by introducing mean shifts in the features, and the model is trained to predict violations based on shifts in the features. Conversely, Fair-MAML focuses on training a model to a new task with minimum data. This is realized through using fairness regularizers for Demographic Parity and Equal Opportunity in the loss function of the MAML framework.

Wang et al. (2022) deals with the issue of fairness in information retrieval systems. It proposes a Meta-learning based Fair Ranking (MFR) model which aims to mitigate bias for protected groups through an

automatically-weighted loss function. It uses a meta learning framework that involves training a meta learner from an unbiased sampled dataset(meta-dataset) while simultaneously training a listwise learning-to-rank(LTR) model on the biased dataset. The meta-learner acts as a weighting function that guides the ranking loss to focus more on the minority groups, thus enhancing fairness. MFR model is formulated as a bilevel optimization problem, where the meta-learner and the ranking model interact to optimize the fairness related parameters.

In Zhao et al. (2021b), a framework termed as Primal-Dual Fair Meta-Learning (PDFM) is designed to enhance fairness in few-shot meta-learning scenarios. PDFM operates by learning suitable primal and dual parameters, thereby enabling the model to quickly adapt to new fairness learning tasks. Unlike MAML, where parameters update takes place in outer loop, in PDFM a pair of primal-dual meta parameters are updated through dual decomposition. In Zhao et al. (2021a), an online meta-learning algorithm FFML has been introduced. FFML learns effective priors for a fair classification model's primal and dual parameters, which govern accuracy and fairness respectively. The problem is formulated using bi-level convex-concave optimization, leading to sub-linear regret and cumulative fairness constraint violation bound.

Although fairness-aware meta learning is a well studied concept, Our work differs from the above discussed existing works in several aspects. For example, Slack et al. (2020) integrates fairness regularizers into the MAML loss function. While the Fair-MAML model discussed in their work might seem similar to ours, it focuses more on K-shot shot learning and does not focuses on individual task treatment based on subpopulations. In contrast Wang et al. (2022) uses a weighted loss function. Our approach stands out by its task specific inner loop optimization, the explicit consideration of subpopulation characteristics during the training process and absence of regularizers. The use of binary cross entropy loss is distinctive to our approach.

## 3 Preliminaries

### 3.1 MAML

Model Agnostic Machine Learning (MAML) Finn et al. (2017) is a popular strategy used in meta-learning. The strategy aims to learn a flexible initialization of a model that can be easily adapted to generalize to many downstream tasks with limited data. Therefore, it is very suitable for few-shot classification tasks. It works on initializing the model parameters so that a small number of gradient steps will help the model learn a new task effectively.

At the core, MAML deals with the problem that machine learning models require large amounts of task-specific data to perform well. On the other hand, humans learn new skills and concepts quickly. For example, a person who has learned to drive a car will find driving other vehicles, such as trucks and buses easier. We also expect machine learning models to behave similarly and learn from a minimum amount of data. MAML avoids this problem by decomposing the model update process into two steps :

- Outer Loop Update (Meta-test task and Meta Update): The outer loop seeks to optimize the model parameters to adapt to new tasks quickly. It is done by finding a suitable initialization of model parameters that perform well on various tasks.

- Inner Loop Update (Meta-train or Task-Specific Update): It can be visualized as having a task-specific learner. The inner loop will use the initialized parameters to adapt to specific tasks using only a few examples from the given task.

The set of data points used in the inner loop is referred to as the support set ($X_s$), and the set of data points used in the outer loop is called the query set ($X_q$). In the inner loop, the loss is computed over task-specific data. This shows how well the model performs on a task using current parameters. The model parameters are then updated using the gradient descent algorithm (i.e., the backpropagation technique in the case of neural networks). This may be performed on a set of tasks, and in the outer loop, the losses across all the tasks are aggregated, and the model parameters are updated using gradient descent. The outer loop's ability

to initialize the parameters and the inner loop's ability to fine-tune them makes MAML suitable for a wide range of tasks where generalization and adaption on a diverse set of tasks are important.

In supervised learning, the meta model, parameterized as $f_\theta$, is trained across tasks $T = \{X, \mathcal{L}\}$ with dataset $X$ and loss function ($\mathcal{L}$). Optimal parameters $\theta^*$ are found by minimizing the following expected loss:

$$\theta^* = \arg\min_\theta \mathbb{E}_{T \sim P(T)}[\mathcal{L}(f_\theta, X)], \tag{1}$$

where $P(T)$ is the distribution over the tasks.

Few-shot meta-learning involves an inner loop for fine-tuning on the support set $X_s$:

$$\theta' = \theta - \eta \nabla_\theta \mathcal{L}(f_\theta, X_s), \tag{2}$$

where $\eta$ is the learning rate.

The outer loop optimizes $\theta$ for improved generalization:

$$\theta^* = \arg\min_\theta \mathbb{E}_{T \sim P(T)}\{\mathcal{L}(f_{\theta - \eta \nabla_\theta L(f_\theta, X_s)}, X_q)\} \tag{3}$$

This approach harnesses multiple tasks to adapt $\theta$, enabling effective generalization to new tasks with limited labeled data. Figure 1 provides a brief overview of the MAML strategy. The green region denotes the inner loop updates on the support set ($X_s$). The purple region denotes the outer loop update over the query set ($X_q$) that acts as a meta-test. In essence, MAML aligns the updates computed during the inner loop on $X_s$ with those computed during the outer loop on $X_q$, enabling better generalization for the model parameters onto the (new) meta tasks (despite a small number of data samples).

Moreover, MAML can learn various complex and challenging tasks by increasing the number of iterations ($I$, in Figure 1) in the inner loop. By performing multiple inner loop updates, the model can generalize better to a specific task and fine-tune its parameters on task-specific data. Another advantage of MAML is that it is a model and task-agnostic algorithm that can be applied to any learning model and algorithm trained using the gradient descent procedure.

### 3.2 Fairness Metrics

Various fairness metrics have been proposed to measure the bias and discrimination in machine learning models quantitatively. Two of the widely used metrics are demographic parity and equalized odds. We used the following metrics to assess the fairness of models trained in this work.

- Demographic Parity (Statistical Parity): It measures whether the positive predictions in a model's output are equally distributed across all the demographic groups. A model is fair if it has a roughly equal positive outcome rate across all the subpopulations. Formally a model satisfies demographic parity if:

$$P(\hat{Y} = 1|A = a) = P(\hat{Y} = 1|A = b)$$

  where $\hat{Y}$ represents the model's prediction and A represents the sensitive attributes for all a, b $\in$ A

- Equalised Odds: It is a fairness metric that evaluates whether the true positive and false negative rates are equal across different demographic groups. In other words, it measures if the model treats all groups equally in correctly classifying positive instances (true positives) and incorrectly classifying negative instances as positive (false negatives). Formally a model satisfies equalized odds if:

$$P(\hat{Y} = 1|Y = i, A = a) = P(\hat{Y} = 1|Y = i, A = b)$$

  where $\hat{Y}$ represents the model's prediction and A represents the sensitive attributes for all a, b $\in$ A and i $\in$ {0, 1} are the targets for a binary classification task.

# 4 Proposed Method

In this section, we describe the methodology employed to improve fairness in the proposed approach. Our approach capitalizes on the unique characteristics of MAML to generalize better onto new tasks, while we employ Multi-Layer Perceptron (MLP) as the underlying learning architecture. The Crux of the proposed approach is the update strategy of the MLP classifier. We divide the training dataset into two subpopulations: protected and unprotected. Because of the inherent bias present in the dataset, they can be treated as two distinct tasks. Via aligning these tasks, we observe that the classifier considers both important and learns to behave similarly towards both.

In MAML, during the inner loop, the model is fine-tuned on the support set, which comprises (multiple) task-specific data. Then in the outer loop, the model parameters are updated based on the losses computed on the query set.

The objective of the MAML-like methods looks as follows:

$$\min_{\theta_0} \mathcal{L}_{\text{outer}}(\arg\min_{\theta_0} \mathcal{L}_{\text{inner}}(\theta_0, X_{\text{s}}), X_{\text{q}}), \tag{4}$$

where $\theta_0$ indicates the initial parameters, $\mathcal{L}_{\text{inner}}$ denotes the task-specific inner loss (meta-train), and $\mathcal{L}_{\text{outer}}$ denotes the outer loss (meta-test), and $X_{\text{s}}$ and $X_{\text{q}}$ denote the inner loop and the outer loop training example sets.

In our approach to improve the fairness, we focus on a single task in the inner loop of MAML framework, executed on $X_s$. The $X_s$ consists of all the protected samples and the $X_q$ consists of all the unprotected samples in the training data. Thus, the objective optimized by our approach is

$$\min_{\theta_0} \mathcal{L}(\arg\min_{\theta_0} \mathcal{L}(\theta_0, X_{\text{s}}), X_{\text{q}}) \tag{5}$$

We perform single or multiple gradient steps based on the configuration (ref section 5.4) using samples from the protected class, on parameters $\theta_0$ to obtain $\theta^i$ as $\theta^i = \theta^{i-1} - \eta \nabla_{\theta^{i-1}} \mathcal{L}(\theta^{i-1}, X_s)$. Then we optimize on the remaining batch samples with the parameters $\theta_i$ and make a final update. Here $\theta^i$ represents the model parameters after the inner loop has finished i iterations.

Note that the subpopulations are divided based on the sensitive feature attributes, and they will have instances of both labels, although skewed in number. The loss function ($\mathcal{L}$) used in both inner and outer loops is binary cross-entropy loss. Note that the inner loop can be run for multiple iterations before executing the outer loop.

## 4.1 Gradient Alignment with meta-learning

Inspired by Patel et al. (2023), this subsection details how the MAML framework aligns the loss gradients computed on the two subpopulations.

One can rewrite equation (5) as

$$\min_{\theta_0} \mathcal{L}(\theta^i, X_{\text{q}}) = \mathcal{L}(\theta_0 - \eta \nabla_{\theta_0} \mathcal{L}(\theta_0, X_s), X_q) \tag{6}$$

$$\frac{\partial \mathcal{L}(\theta^i, Xq)}{\partial \theta_0} = \nabla \mathcal{L}(\theta^i, X_q) \cdot \frac{\partial \theta^i}{\partial \theta_0} \text{(chain rule of differentiation)}$$

$$= \nabla \mathcal{L}(\theta^i, X_q) \cdot \frac{\partial(\theta_0 - \eta \nabla \mathcal{L}(\theta_0, X_s))}{\partial \theta_0}$$

$$= \nabla \mathcal{L}(\theta^i, X_q).(I - \eta \nabla^2 \mathcal{L}(\theta_0, X_s))$$

$$\approx (\nabla \mathcal{L}(\theta_0, X_q) - \nabla^2 \mathcal{L}(\theta_0, X_q) \cdot \eta \nabla \mathcal{L}(\theta_0, X_s))(I - \eta \nabla^2 \mathcal{L}(\theta_0, X_s))$$

(Taylor series approx. of $\nabla \mathcal{L}(\theta^i, X_q)$ in the neighborhood of $\theta_0$)

$$\approx \nabla \mathcal{L}(\theta_0, X_q) - \eta \nabla^2 \mathcal{L}(\theta_0, X_q) \cdot \nabla \mathcal{L}(\theta_0, X_s) - \eta \nabla^2 \mathcal{L}(\theta_0, X_s) \nabla \mathcal{L}(\theta_0, X_q)$$

(Given that $\eta$ is small, higher terms involving $\eta^2$ are negligible)

$$= \nabla \mathcal{L}(\theta_0, X_q) - \eta \nabla \left( \nabla \mathcal{L}(\theta_0, X_q) \cdot \nabla \mathcal{L}(\theta_0, X_s) \right)$$

(using the product rule $\nabla a{\cdot}b + \nabla b{\cdot}a = \nabla(a{\cdot}b)$)

Observing the above gradient equation closely reveals that for reducing $\mathcal{L}(\theta^i, X_q)$, the overall gradients would minimize $\mathcal{L}(\theta_0, X_q)$ while maximizing the product $\nabla \mathcal{L}(\theta_0, X_s) \cdot \nabla \mathcal{L}(\theta_0, X_q)$. In other words, optimizing for equation (6) causes a regularization effect to align $\nabla \mathcal{L}(\theta_0, X_s)$ and $\nabla \mathcal{L}(\theta_0, X_q)$. This avoids any subpopulation strongly driving the learning process, reducing bias in the resulting model.

## 5 Experiments

### 5.1 Datasets

To evaluate the proposed framework empirically, we use the following four real-world datasets.

1. **Compas Dataset** ProPublica (2013): contains information about the criminal defendants who have been arrested from Broward Country, Florida. The dataset is used for learning models that predict the likelihood of re-offending based on attributes such as gender, race, prior criminal history, and risk scores. We consider a binary sensitive feature whether a criminal is African-American or not.

   The Compas dataset has been widely used in research but is subject to criticism because of inherent biases in the data affecting minority groups. Many studies ( Rudin et al. (2020), Bao et al. (2022)) raise concerns about the ethical implications of using this dataset. Despite these issues, Compas dataset remains relevant in fairness research because of it's widespread use. However, caution is advised when interpreting results from this dataset due to its known limitations.

2. **Adult Dataset** Becker & Kohavi (1996): contains $48,842$ instances, each characterized by 14 attributes. These attributes provide insights into the demographic and socioeconomic status of subjects such as age, education, occupation, and marital status. The dataset includes a binary label that classifies an individual's income level into two categories: above or below $\$50,000$ per year. We consider gender, i.e. male and female as the sensitive feature attribute where females are regarded as the protected subgroup.

3. **Communities and Crime Dataset** Redmond (2011): aims to predict the crime rates using attributes of communities in the United States including educational level, unemployment rates, poverty rates, and race. The predicted variable violent crime rate is a continuous value. To facilitate our analysis, we convert it into a binary label based on whether the community is in the top 50% violent crime rate within the state Slack et al. (2020). Additionally, We include a binary sensitive column that receives a protected label (1) if African-Americans constitute the highest or second highest percentage of the community's racial makeup.

4. **German Credit Dataset** Hofmann (1994): contains data of 1000 loan applicants with each applicant being classified as "good" or "bad" credit risk based on various attributes such age, sex, job

and credit history. To examine the potential biases in loan approval Sex is considered as a sensitive attribute. Approximately 30% females in the dataset form the protected group and 70% males form the unprotected group.

The choice of these datasets was driven by their popularity among recent fairness literature and the valuable diversity they contribute, reflecting variations in social settings, sensitive features, and class imbalance. Despite their apparent similarities, these datasets differ in sensitive features ranging from race and gender to income levels. This diversity in sensitive features is critical for our evaluation, highlighting the framework's robustness across various real-world scenarios.

## 5.2 Baseline Models

To facilitate comparison, we establish baseline models for each of the datasets using a Multi-Layer Perceptron (MLP) with a single hidden layer. The baseline is constructed without the use of the MAML framework. A fixed learning rate of 0.001 is utilized, and the ReLU activation function is applied during training. In Table 1, we present a comprehensive overview of the accuracy and fairness metrics obtained from the baseline models across all four datasets. It is observed from the baseline model results (first row of each horizontal panel in Table 1) that the accuracy values are notably high. However, a closer look at fairness metrics indicates the high values of Demographic Parity Difference (DP diff) and Equalized odds Difference (EOD Diff). This indicates the presence of bias in the model particularly towards the minority group and highlights the importance of addressing the fairness concerns.

- **MBA for DP and EOD:** We also compare the proposed algorithm with a manually adjusted baseline for Demographic Parity Difference ( Kamiran & Calders (2011)) and Equalized Odds Difference ( Hardt et al. (2016))(refer second and thirds rows of each dataset in Table 1). For the manual adjustment of DP diff, equal representation of subpopulations in the training data is ensured using oversampling from the protected group. This is done to handle class imbalance, with the goal of improving fairness by ensuring that both protected and unprotected groups are equally represented during training. To manually adjust the baseline for better EOD, we ensure that the True Positive Rate (TPR) and False Positive Rate (FPR) are the same across the sensitive groups. Misclassified samples in each group are rebalanced to equalize these rates, making sure that the model's error rates do not disproportionately affect the fairness of one group more than the other.

- **Threshold Adjustment:** Additionally, a threshold-based baseline was implemented to optimize demographic parity difference, where different classification thresholds ranging from 0.1 to 1.0, with a step size of 0.1, were applied for the protected group to further adjust and improve fairness metrics. The corresponding DP diff values and accuracies are plotted in 2. However, since the optimization is focused on DP diff, the Equalized Odds diff values are relatively poor. In the third row of each dataset in Table 1, we include the EOD Difference, accuracy, and DP Difference for the threshold that minimizes the DP Difference.

While we present three manually adjusted baselines, it is important to note that multiple baselines could be constructed to optimize either DP diff, EOD diff, or both, based on the specific fairness objective being pursued.

## 5.3 Experimental Setup

The datasets at hand not only differ in terms of complexity but also in size and underlying nature of bias. Because of this, we use slightly different model architectures and hyper-parameters such as activation functions, and learning rates of the inner and outer loops across different datasets.

The datasets were divided into training, validation and test sets in a 70%, 15%, 15% split respectively. We optimized the baseline models for accuracy and the selected hyperparameters were used for subsequent training of the models. We explored various combinations of inner and outer learning rates to identify the most effective settings. During training and validation, the sensitive features were excluded from the model

Table 1: Comparative analysis of results obtained across the four datasets over 5 independent runs. Results are presented varying the number of inner loop iterations for both the proposed configurations. Arrows (↑, ↓) next to the metrics indicate the direction of better performance (more is better and less is better, respectively). Bold values indicate the best performances. Please note that the threshold adjustment is primarily aimed at optimizing the DP difference, though in some cases it may also incidentally improve the EO difference. (Here Prot. represents protected samples and Unprot. represents unprotected samples and Manual Baseline Adjustment (MBA) refers to the manual tuning of fairness baselines)

| Dataset | $X_s$ and $X_q$ | No. of inner iterations | Accuracy (↑) | DP diff (↓) | EOD diff (↓) |
|---|---|---|---|---|---|
| **Compas Dataset** | Baseline Models | - | **67.37** | 0.24 | 0.25 |
| | MBA for DP | - | 67.86 | 0.26 | 0.26 |
| | MBA for EOD | - | 63.67 | 0.16 | 0.17 |
| | Threshold adjustment | - | 56.27 | 0.06 | 0.11 |
| | Config. 1 | 1 | $59.33 \pm 0.02$ | $0.07 \pm 0.06$ | $0.07 \pm 0.05$ |
| | | 10 | $60.21 \pm 0.02$ | $0.06 \pm 0.02$ | $0.07 \pm 0.02$ |
| | Config. 2 | 1 | $57.55 \pm 0.02$ | $\mathbf{0.04 \pm 0.05}$ | $\mathbf{0.04 \pm 0.04}$ |
| | | 10 | $59.30 \pm 0.02$ | $\mathbf{0.04 \pm 0.03}$ | $\mathbf{0.04 \pm 0.03}$ |
| **Adult Dataset** | Baseline Models | - | **82.86** | 0.21 | 0.30 |
| | MBA for DP | - | 82.59 | 0.20 | 0.25 |
| | MBA for EOD | - | 83.02 | 0.12 | 0.10 |
| | Threshold adjustment | - | 80.19 | 0.05 | 0.34 |
| | Config. 1 | 1 | $67.05 \pm 0.07$ | $0.08 \pm 0.05$ | $0.11 \pm 0.05$ |
| | | 10 | $52.58 \pm 0.07$ | $0.1 \pm 0.06$ | $\mathbf{0.07 \pm 0.04}$ |
| | Config. 2 | 1 | $64.58 \pm 0.06$ | $\mathbf{0.07 \pm 0.07}$ | $\mathbf{0.07 \pm 0.12}$ |
| | | 10 | $60.57 \pm 0.08$ | $0.09 \pm 0.04$ | $0.12 \pm 0.05$ |
| **Communities & Crime Dataset** | Baseline Models | - | **70.22** | 0.14 | 0.12 |
| | MBA for DP | - | 73.58 | 0.16 | 0.10 |
| | MBA for EOD | - | 70.50 | 0.13 | 0.10 |
| | Threshold adjustment | - | 49.62 | 0.01 | 0.01 |
| | Config. 1 | 1 | $59.80 \pm 0.03$ | $\mathbf{0.06 \pm 0.04}$ | $\mathbf{0.06 \pm 0.03}$ |
| | | 10 | $63.25 \pm 0.06$ | $0.07 \pm 0.08$ | $0.08 \pm 0.11$ |
| | Config. 2 | 1 | $58.59 \pm 0.03$ | $\mathbf{0.06 \pm 0.06}$ | $0.07 \pm 0.05$ |
| | | 10 | $62.00 \pm 0.08$ | $0.08 \pm 0.04$ | $\mathbf{0.06 \pm 0.05}$ |
| **German Credit Dataset** | Baseline Models | - | **71.47** | 0.15 | 0.18 |
| | MBA for DP | - | 71.4 | 0.11 | 0.17 |
| | MBA for EOD | - | 72.40 | 0.11 | 0.15 |
| | Threshold adjustment | - | 70.0 | 0.01 | 0.07 |
| | Config. 1 | 1 | $70.70 \pm 0.04$ | $0.04 \pm 0.04$ | $0.04 \pm 0.03$ |
| | | 10 | $70.00 \pm 0.05$ | $0.03 \pm 0.01$ | $0.07 \pm 0.02$ |
| | Config. 2 | 1 | $70.67 \pm 0.01$ | $0.03 \pm 0.03$ | $0.04 \pm 0.03$ |
| | | 10 | $71.00 \pm 0.01$ | $\mathbf{0.02 \pm 0.01}$ | $\mathbf{0.04 \pm 0.02}$ |

to prevent any bias in the model learning. However, the sensitive features were incorporated during the inference phase to ensure that the performance metrics consider these critical attributes. For the compas dataset, an MLP with one hidden layer, the ReLU activation function, and inner and outer learning rates of 0.01 were used. For the Adult dataset, also similar setup with two hidden layers was used, and the inner and outer loop learning rates were 0.001. For the Communities and Crime dataset, MLP with one hidden layer, tanh activation function, and inner and outer loop learning rates of 0.005 and 0.001 were used. For the German Credit Dataset, MLP with one hidden layer, relu activation function and inner and outer learning rates of 0.005 and 0.1 were used. We use the fairlearn library Weerts et al. (2023) to compute the mentioned fairness metrics.

Table 2: Comparative analysis of results obtained from all four datasets averaged over five independent runs, exploring the impact of different proportions of protected and unprotected samples within the support set ($X_s$) and the query set ($X_q$)

| Dataset | Num. of inner iter. | $X_q : X_s$ | Accuracy (↑) | DP diff (↓) | EOd diff (↓) |
|---|---|---|---|---|---|
| **compas Dataset** | 1 | 80 : 20 | **64.45 ± 0.04** | 0.16 ± 0.09 | 0.17 ± 0.10 |
| | | 70 : 30 | 63.80 ± 0.04 | 0.16 ± 0.09 | 0.17 ± 0.09 |
| | | 60 : 40 | 59.69 ± 0.03 | 0.07 ± 0.06 | 0.08 ± 0.06 |
| | | 50 : 50 | 59.69 ± 0.03 | 0.05 ± 0.03 | 0.05 ± 0.02 |
| | 10 | 80 : 20 | 59.88 ± 0.03 | 0.06 ± 0.04 | 0.06 ± 0.04 |
| | | 70 : 30 | 61.02 ± 0.04 | 0.08 ± 0.06 | 0.08 ± 0.06 |
| | | 60 : 40 | 59.51 ± 0.03 | 0.05 ± 0.03 | **0.04 ± 0.02** |
| | | 50 : 50 | 59.43 ± 0.02 | **0.04 ± 0.02** | **0.04 ± 0.02** |
| **Adult Dataset** | 1 | 80 : 20 | **75.53 ± 0.01** | 0.08 ± 0.07 | 0.08 ± 0.07 |
| | | 70 : 30 | 70.75 ± 0.05 | **0.07 ± 0.05** | 0.09 ± 0.05 |
| | | 60 : 40 | 64.58 ± 0.07 | **0.07 ± 0.05** | **0.07 ± 0.05** |
| | | 50 : 50 | − | − | − |
| | 10 | 80 : 20 | 63.67 ± 0.09 | 0.13 ± 0.08 | 0.13 ± 0.09 |
| | | 70 : 30 | 52.89 ± 0.11 | 0.10 ± 0.07 | **0.07 ± 0.05** |
| | | 60 : 40 | 52.58 ± 0.07 | 0.10 ± 0.06 | **0.07 ± 0.04** |
| | | 50 : 50 | − | − | − |
| **Communities and crime Dataset** | 1 | 80 : 20 | 62.45 ± 0.02 | 0.08 ± 0.05 | 0.09 ± 0.01 |
| | | 70 : 30 | **63.10 ± 0.02** | 0.1 ± 0.04 | 0.07 ± 0.04 |
| | | 60 : 40 | 56.94 ± 0.03 | **0.05 ± 0.05** | 0.06 ± 0.03 |
| | | 50 : 50 | 59.80 ± 0.03 | 0.06 ± 0.04 | 0.06 ± 0.03 |
| | 10 | 80 : 20 | 53.78 ± 0.03 | **0.05 ± 0.04** | **0.05 ± 0.03** |
| | | 70 : 30 | 52.78 ± 0.03 | 0.09 ± 0.03 | 0.12 ± 0.07 |
| | | 60: 40 | 52.93 ± 0.06 | 0.07 ± 0.08 | 0.10 ± 0.011 |
| | | 50 : 50 | 52.83 ± 0.06 | 0.06 ± 0.08 | 0.07 ± 0.09 |
| **German Credit Dataset** | 1 | 80 : 20 | **71.29 ± 0.04** | 0.04 ± 0.04 | 0.05 ± 0.05 |
| | | 70 : 30 | 69.57 ± 0.05 | 0.04 ± 0.04 | 0.09 ± 0.07 |
| | | 60 : 40 | 70.70 ± 0.04 | 0.04 ± 0.04 | **0.04 ± 0.03** |
| | | 50 : 50 | - | - | - |
| | 10 | 80 : 20 | 69.17 ± 0.03 | 0.03 ± 0.02 | 0.07 ± 0.04 |
| | | 70 : 30 | 70.70 ± 0.04 | 0.03 ± 0.02 | 0.06 ± 0.05 |
| | | 60: 40 | 70.00 ± 0.05 | **0.03 ± 0.01** | 0.07 ± 0.02 |
| | | 50 : 50 | - | - | - |

The experiments were conducted on a workstation with an *Intel(R) Core(TM) i7-9750H* CPU running at 2.60GHz, which features 6 physical cores and 12 logical threads. Additionally, the system was equipped with an NVIDIA GeForce GTX 1650 GPU and 32 GB of RAM. The experiments were performed on a Windows 10 operating system. The software environment utilized Python version 3.11.1 for implementing and running the experiments. The longest training time required for the proposed method was less than 5 minutes, and the inference time was instantaneous.

## 5.4 Results and Analysis

We present a couple of variants (of the support and query sets) in line with our proposed methodology. Note that the flexibility of the MAML framework to distribute tasks between the inner and outer loops can be exploited to accommodate different variants. However, we present the following two as very intuitive configurations.

Table 3: Accuracy results for protected and unprotected groups on the compas dataset for varying numbers of inner iterations and different splits of $X_q$ and $X_s$

| Number of inner iterations | $X_q : X_s$ | Protected Accuracy | Unprotected Accuracy |
|---|---|---|---|
| | Baseline model | 0.6682 | 0.7081 |
| 1 | 80:20 | 0.6246 | 0.6706 |
| 1 | 70:30 | 0.623 | 0.6571 |
| 1 | 60:40 | 0.5871 | 0.6554 |
| 1 | 50:50 | 0.5922 | 0.6416 |
| 10 | 80:20 | 0.5767 | 0.6326 |
| 10 | 70:30 | 0.5904 | 0.6427 |
| 10 | 60:40 | 0.5788 | 0.6495 |
| 10 | 50:50 | 0.5888 | 0.6405 |

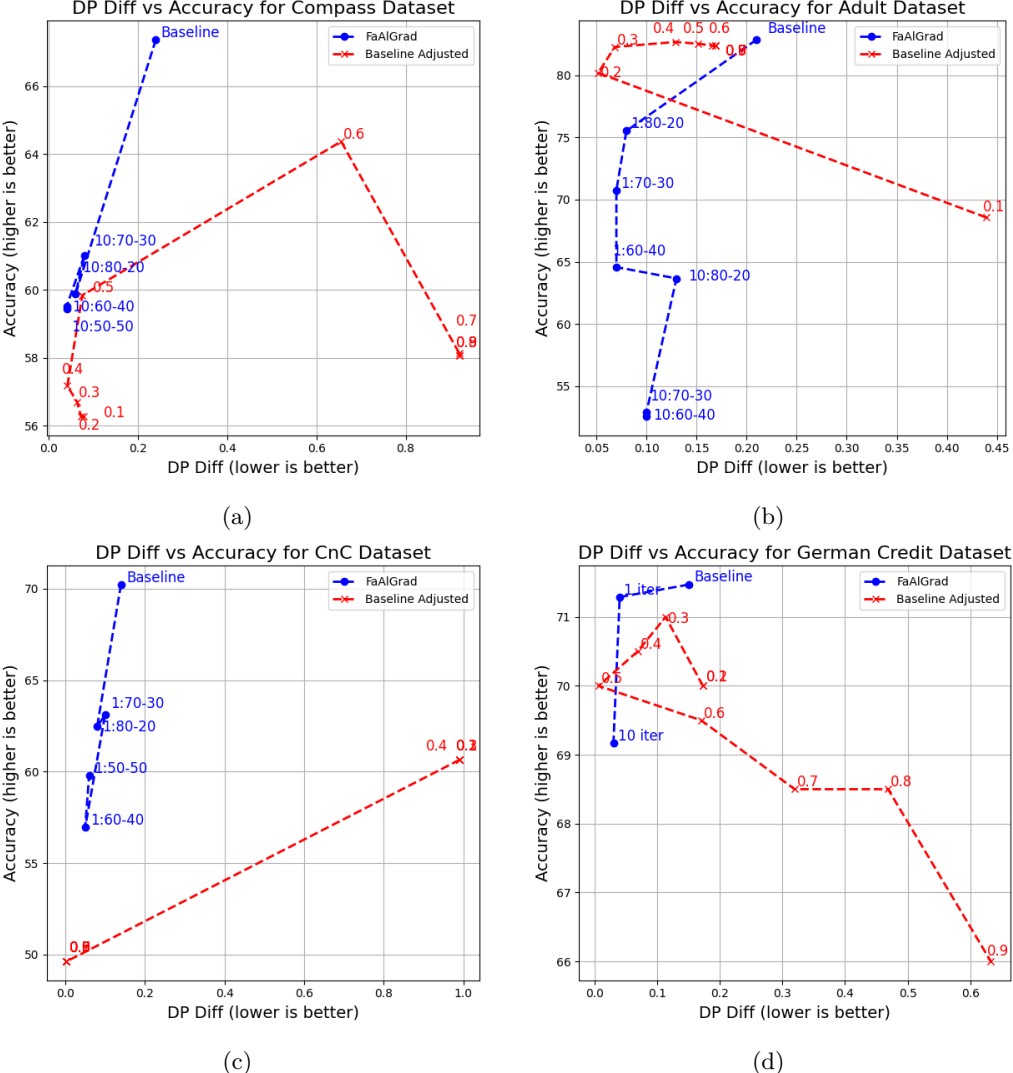

Figure 2: Trade-off plots illustrating the relationship between accuracy and demographic parity difference across various datasets, comparing the proposed method with the baseline using adjusted threshold.

1. **Configuration** 1 **:** In the initial setup, we curate $X_s$ with all the instances from the protected group (e.g., African-Americans in the case of the compas dataset and female gender in the Adult dataset). Correspondingly, $X_q$ is formulated with all the instances from the unprotected group. The This design facilitates the alignment of loss gradients for both the subpopulations, thus enabling the model to effectively counter the bias (refer to sec. 4).

2. **Configuration** 2 **:** In the second setup, the configuration seeks to counteract the way in which the baseline model learns the bias. For instance, in the compas dataset where label 1 corresponds to the person getting rearrested in the next two years, whereas label 0 indicates no rearrest in the subsequent years. From the predictions and fairness behavior of baseline models, it is evident that the minority group i.e. African-Americans, are more likely to be predicted to re-offend as compared to other groups. They are more likely to be predicted positively than Caucasians. Hence, in $X_s$ we assemble all the instances of individuals from the protected group (African-Americans) with ground truth label 0 and instances of the unprotected group with ground truth label 1. Opposite to this, in $X_q$, we keep all the instances of the protected group with ground label 1 and the samples of the unprotected group with ground label 0. In this configuration, the alignment factor as the dot product between the gradients of query and support set (refer to sec. 4.1) is important because it ensures that the bias-reducing updates from the inner loop remain effective when the model is tested on a query set that would typically reinforce bias. It encourages the MAML framework to fine-tune its prediction contrary to prevailing bias.

   Similar adaptations are made for the Communities and Crime dataset; protected samples with ground truth label 0 and unprotected samples with ground truth label 1 comprise $X_s$, whereas the protected samples with ground truth label 1 and unprotected samples with ground truth label 0 are included in $X_q$. There is a slight modification in the Adult dataset. Since gender is chosen as the sensitive attribute, the male is the privileged group with a 31% probability of having a positive outcome ($> \$50K$) compared to an 11% probability of having a positive outcome for the female group. Hence, we keep protected samples (Females) with ground truth 1 and unprotected samples with ground truth 0 in the $X_s$. $X_q$ includes protected samples with ground truth label 0 and unprotected samples with ground truth label 1. Similar setup is used to design $X_s$ and $X_q$ in German Credit Dataset.

The results for both these configurations are presented in Table 1 (rows 5 and 6 in each of the horizontal panels). Clearly, the results show substantial improvement in fairness metrics compared to the baseline models. For the compas dataset, Configuration 1 brings about a noteworthy improvement in the Demographic Parity Difference (DP Diff), reducing it from 0.24 in the baseline to 0.06. An even more impressive DP Diff of 0.04 is achieved in Configuration 2. A parallel trend is observed in the Equalized Odds Difference (EOD Diff), which undergoes a substantial decrease from 0.25 in the baseline to 0.07 in Configuration 1 and 0.04 in Configuration 2.

In the context of the Adult Dataset, the baseline DP Diff of 0.21 experiences a substantial reduction to 0.08 in Configuration 1, further decreasing to 0.07 in Configuration 2. Simultaneously, the EOD Diff improves from 0.30 in baseline models to 0.07 in Configuration 1 and Configuration 2.

The Communities and Crime dataset also shows similar trends, with the DP difference improving from 0.14 in the baseline model to 0.06 in configuration 1 and configuration 2. Correspondingly, the EOD Difference improves from 0.12 in the baseline model to 0.06 in configuration 1 and configuration 2.

For the German Credit Dataset, the DP difference improves to a minimum of 0.2 in configuration 2 as compared to 0.15 in the baseline. Similarly, the EOD difference also reduces from 0.18 in the baseline to 0.04 in configuration 2 with 10 inner iterations.

The results in Table 1 show the effectiveness of both configurations in improving fairness across the four datasets. However, there is a decrease in accuracy. Substantial advancements in fairness metrics counter-balance this decrease. Importantly, these improvements are achieved without resorting to exclusive fairness regularizers (optimizing the fairness metrics) or any of the pre-processing techniques. This observation high-

Table 4: Results on compas dataset for one inner loop iteration when the $X_s$ comprises unprotected samples and $X_q$ comprises protected samples.

| $X_q : X_s$ | Accuracy ($\uparrow$) | DP diff ($\downarrow$) | EOD diff ($\downarrow$) |
|---|---|---|---|
| 80 : 20 | 56.57 | 0.09 | 0.10 |
| 70 : 30 | 57.18 | 0.09 | 0.10 |
| 60 : 40 | 55.58 | 0.06 | 0.08 |
| 50 : 50 | 55.32 | 0.06 | 0.08 |

lights the inherent potential of the proposed gradient alignment approach in addressing bias and promoting fairness in machine learning models.

## 5.5 Varying proportions of $X_s$ and $X_q$

In light of the notable improvements achieved in enhancing fairness metrics through Configuration 1 and Configuration 2 (refer to sec. 5.4), addressing the trade-off between fairness and accuracy is crucial. While these configurations yield remarkable fairness improvements, there is a concurrent reduction in accuracy. We undertake the following exploration to strike a balance between these two essential metrics.

Here, we opt for a distinct setup involving a controlled split of the training data between the two subsets: $X_s$ and $X_q$. The idea here is to vary the ratio of protected samples within $X_s$, while accommodating the remaining protected samples and all unprotected samples within $X_q$. This ratio manipulation occurs within the $80 - 20$ to $50 - 50$ range. This allows us to navigate the trade-off spectrum between fairness and accuracy and choose a desired point of operation.

However, it's imperative to note that the dataset's inherent composition constrains the ability to reduce the ratio beyond $50 - 50$, as protected samples always constitute less than $50\%$ of the total data. This is especially pertinent in the Adult and German Credit dataset, where the limited number of protected samples precludes the achievement of a $50 - 50$ split.

Table 2 provides insights into the interplay between fairness and accuracy within different configurations. Remarkably, it is evident that the highest accuracy values are consistently associated with an $80 - 20$ split between $X_q$ and $X_s$. This is because most of the training data lies in the outer loop in the form of $X_q$, making the model prone to dominances similar to the baseline models.

Moreover, the ablation study reveals a discernible pattern: an increase in the proportion of protected samples within $X_s$ corresponds to an improvement in fairness metrics. Note that this trend may not be strictly linear due to the saturation level in the metrics values (leads to only small numerical differences after the enhancements). However, the general trajectory supports our hypothesis that strongly aligning the loss gradients between the subpopulations leads to a fairer classification model.

Based on the experimental outcomes, classification on the compas dataset demonstrates a peak accuracy of 64.45 at one end, along with an impressive Demographic Parity Difference (DP Diff) and Equalized Odds Difference (EOD Diff) of 0.04 at the other. For the Adult dataset, the operating range spans between a peak accuracy of 75.53 and fairness metrics of 0.07 DP Diff and EOD Diff. Similarly, the highest accuracy observed for the Communities and Crime dataset is 63.10, with the best DP Diff and an EOD Diff of 0.05.

The results also show how increasing the number of inner iterations improves the model's fairness. These outcomes display significant enhancements in fairness metrics compared to baseline models. Most significantly, the results from Table 2 unveil a crucial strategic insight: by varying the ratio of $X_q$ to $X_s$, we can dynamically achieve a desirable equilibrium between fairness and accuracy. This flexibility empowers us to find an operable point with the desired accuracy and fairness behavior without altering other hyper-parameters.

Table 5: Performance on real datasets. Metrics with ↑ indicate the larger the better, and ↓ indicates the smaller the better. Best performances are labeled in bold. The empty entries indicate that the referenced did not work with the corresponding dataset.(EODR refers to the Equalized Odds Ratio and DPR represents the Demographic Parity Ratio)

| | compas | | | Adult | | | Communities and Crime | | |
|---|---|---|---|---|---|---|---|---|---|
| | Acc. (↑) | DPR (↑) | EODR (↑) | Acc. (↑) | DPR (↑) | EODR (↑) | Acc. (↑) | DPR (↑) | EODR (↑) |
| Fair-MAML Slack et al. (2020) | 58 | 0.81 | - | - | - | - | **75.4** | 0.66 | - |
| m-FTML Finn et al. (2019) | - | - | - | **67.91** | 0.54 | 0.43 | 48.69 | 0.38 | 0.29 |
| FFML Zhao et al. (2021a) | - | - | - | 61.35 | **0.91** | 0.87 | 59.57 | 0.74 | 0.69 |
| FaAlGrad (ours) | **59.43** | **0.95** | **0.96** | 52.58 | 0.86 | **0.90** | 59.8 | **0.87** | **0.85** |

## 5.6 Swapping the subpopulations

After demonstrating the effect of varying the proportion of protected samples in $X_s$, we now discuss swapping the subpopulations within the Model-Agnostic Meta-Learning (MAML) framework. In other words, the number of unprotected samples in $X_s$ was varied, and $X_q$ encompassed the remaining unprotected samples and all the protected samples. As the split of $X_s$ increases, more unprotected samples are kept in the inner loop, and a decrease in accuracy and an improvement in fairness metrics is observed. Table 4 contains the results after swapping the subpopulations for compas dataset.

This observed trend is consistent with the inferences from our prior experiments (refer to sec. 5.5) despite swapping protected and unprotected samples within the MAML framework. This observation can be attributed to the fundamental concept of gradient alignment inherent to the proposed MAML-inspired training framework (refer to sec. 4.1). This happens because of aligning of loss gradients of the inner loop and outer loop in the MAML framework. The model tries to learn the common features between $X_s$ and $X_q$, making the learning independent of the sensitive feature attributes. This separation of subpopulations i.e. protected and unprotected groups, helps the model to behave fairly via gradient alignment.

## 5.7 Comparative Results

The effectiveness of our proposed approach is evaluated through a comprehensive comparison with other state-of-the-art models and methodologies that employ the Meta-Learning framework to enhance fairness in machine learning models. The models for this comparison include:

- Fair MAML (DP regularized) Slack et al. (2020)

- m-FTML  Finn et al. (2019)

- FFML  Zhao et al. (2021a)

Table 5 presents the accuracy and fairness metrics for comparison. Upon careful evaluation of the results, the proposed method, FaAlGrad, demonstrates notable superiority across all four datasets compared to the aforementioned meta-learning-driven works for improving fairness. Particularly striking improvements are observed in the context of the Communities and Crime dataset. To facilitate a meaningful comparison, we have selected the best-performing split between $X_s$ and $X_q$ for each dataset from Table 2. This ensures a fair and relevant performance assessment against existing approaches wherein EOD Ratio and DP Ratio have been used for comparison.

### 5.8 Demonstrating Bias Reduction in FaAlGrad

In this subsection, we demonstrate the reduction in bias achieved by the proposed FaAlGrad algorithm using the feature attribution technique. Specifically, we use the SHAP Lundberg & Lee (2017) graphs to interpret the reduction in bias. SHAP values provide the contribution of each feature to the classifier's prediction. In the context of fairness, SHAP values will help us find how different features (especially sensitive ones such as gender, race, etc.) affect the model prediction. SHAP graph visualizes the impact of individual features on the model's prediction for a specific data sample. A positive SHAP value indicates that the feature contributes to a higher prediction (favors the positive class), whereas a negative value indicates that the feature contributes to a lower prediction (favors the negative class). To assess the effectiveness of our algorithm, we generated SHAP graphs for both baseline models and models trained using the FaAlGrad algorithm on various datasets. We chose beeswarm plots to present the SHAP values visually. The SHAP graphs were generated during the inference phase, as that is the only phase when sensitive attributes were passed to the model.

Figure 3a and Figure 3b represent the SHAP graphs for the baseline model and the model trained using FaAlGrad, respectively, for the compas dataset. The features are ranked from top to bottom by their mean absolute SHAP values in the graph. The points are distributed horizontally across the x-axis based on their SHAP values. One may specifically focus on the contribution of sensitive features 'race' on the model's prediction. One can observe that in Figure3a, higher values of the sensitive feature (protected group) have negative SHAP values, and lower values of the sensitive feature (unprotected group) have positive SHAP values. This indicates that the protected group (African Americans) has a higher likelihood of re-offending. This reflects the bias in the model's prediction. However, Figure 3b demonstrates the impact of FaAlGrad. Notice the difference in the scale of the x-axis. Higher values of the sensitive feature now cluster closely to the y-axis equally on both sides of the graph. This shows that the sensitive feature 'race' now impacts positive and negative outcomes equally. This indicates a significant reduction of bias by the model. Therefore, analyzing the SHAP graphs provides more evidence (apart from the metrics) for the bias reduction achieved by our FaAlGrad algorithm.

We extend this analysis to the Adult dataset illustrating the algorithm's consistent performance in mitigating the bias. Figure 4a and 4b are the SHAP graphs for the baseline and FaAlGrad algorithm trained on the Adult dataset, respectively. Here, the sensitive feature is gender, and one may focus primarily on its contribution to the model's prediction. We see that in Figure4a, higher values of the sensitive feature (refers to the protected group, females) have negative SHAP values, and lower values of the sensitive feature (refers to the unprotected group, males here) have positive SHAP values. This indicates that protected group (females) have higher chances of being predicted as having salaries less than $50,000 per year as compared to unprotected group (males). This reflects the bias in the model's prediction. However in Figure 4b, we notice that higher as well as lower values of the sensitive feature now cluster closely to the y-axis equally on both sides of the graph. This shows that the sensitive feature of gender now impacts positive and negative outcomes almost equally. This indicates that the model is no longer biased toward the protected group.

Also, for the baseline model, it is observed that sensitive feature gender is ranked at the top, indicating its significant impact on the model output. After applying FaAlGrad the feature gender shifts to bottom demonstrating a change in feature's importance. This shift indicates a reduction in the influence of the sensitive feature on the model's prediction, implying fairness enhancement of the model. Similar results were obtained when analyzing the algorithm on the Communities and Crime dataset. However, due to the large number of input features, we are unable to present the results here.

## 6 Conclusion

In this paper, we presented a meta-learning inspired training framework for alleviating the bias in classifiers. The core idea of the approach is to handle the dominance caused by certain subpopulations in the training data. The proposed approach demonstrates the ability of the MAML framework to naturally facilitate the computation of loss gradients subpopulation-wise and align them. To the best of our knowledge, this is not reckoned by any of the existing meta-learning-driven works Finn et al. (2019); Slack et al. (2020); Zhao et al.

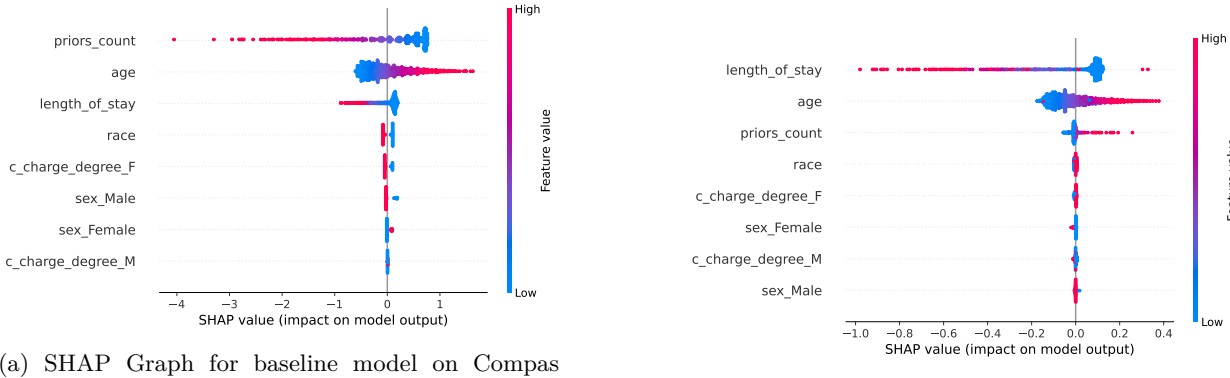

(a) SHAP Graph for baseline model on Compas Dataset

(b) SHAP Graph for FaAlGrad on Compas Dataset

Figure 3: Comparison of SHAP graphs for different models on the compas Dataset

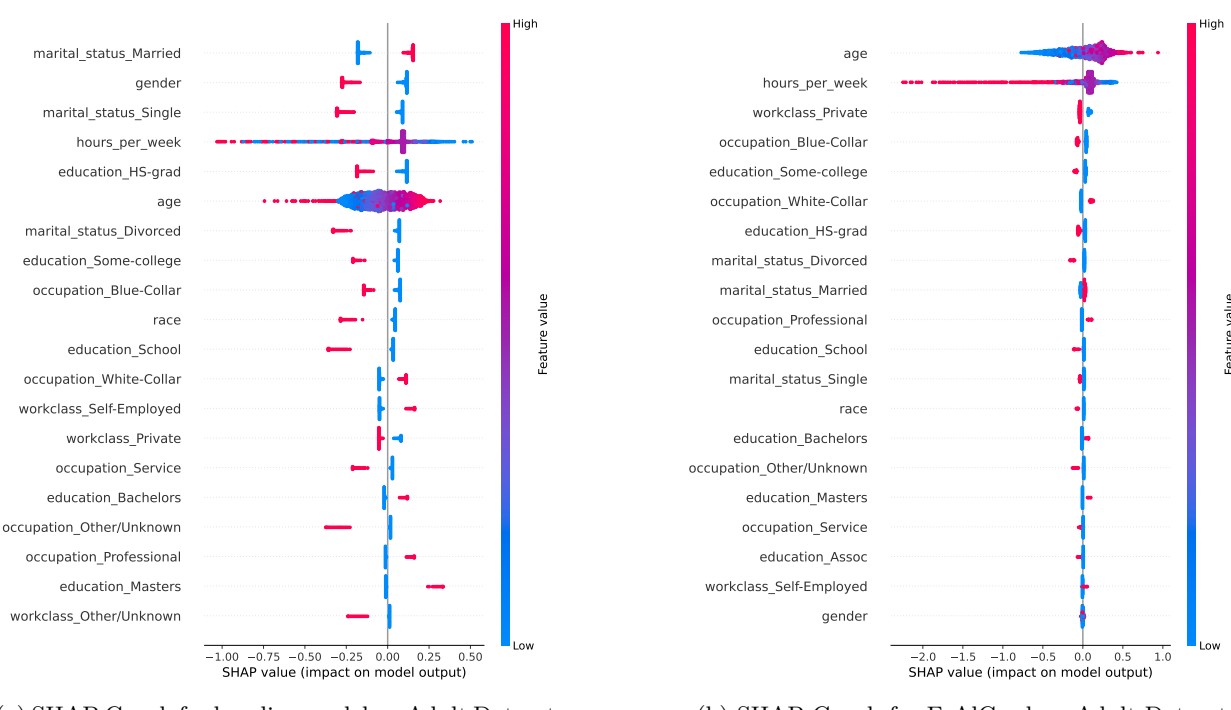

(a) SHAP Graph for baseline model on Adult Dataset

(b) SHAP Graph for FaAlGrad on Adult Dataset

Figure 4: Comparison of SHAP graphs for different models on the Adult Dataset

(2021a) toward improving fairness. Moreover, the ablations reveal that the framework provides a range of operating points to pick a desired system characteristic.

Without adding any exclusive fairness regularizers (in the form of the discussed metrics), optimizing only the cross-entropy loss significantly improved the fairness of the classifiers. We plan to investigate the behavior of the proposed 'gradient alignment' regularization in greater detail, particularly by adding more direct fairness regularizers, extending to more complex tasks such as vision datasets, realizing it outside the meta-learning framework, etc. Further, in contrast to many other approaches that heavily rely on preprocessing, our proposed method eliminates the need for any preprocessing.

It's important to note that although the experiments may compromise on accuracy, this work represents a novel approach towards bias reduction by leveraging multiple tasks within MAML. The findings demonstrate considerable potential for leveraging these tasks to implement bias reduction effectively.

## Acknowledgment

We sincerely thank Dr. Manisha Padala, IIT Gandhinagar for her valuable discussions during our experimental setup, which greatly contributed to shaping this work.

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
