# OpenReview forum: "FaAlGrad: Fairness through Alignment of Gradients across Different Subpopulations"
_TMLR — Accepted by TMLR_

### Review · Reviewer_4uSp · 2024-08-16

**Summary Of Contributions:**

This paper discusses the use of MAML in a fairness setting, with one sensitive attribute corresponding to the support set and the other to the query set. The authors argue that the gradient alignment induced by MAML is helpful for fairness objectives. Empirically, they demonstrate that this approach is able to improve fairness metrics on some standard datasets, at the cost of some accuracy, and demonstrate that the SHAP values from the model after this method is applied put less importance on sensitive features.

**Audience:**

Yes

**Broader Impact Concerns:**

I generally advise against the use of the COMPAS dataset in fairness papers that do not explicitly consider the criminal justice context. See “It’s COMPASlicated” (Bao et al) for more details on this. I’d recommend at the very least a significant disclaimer about the issues involved in this dataset

**Claims And Evidence:**

No

**Requested Changes:**

See critiques above for details. I think the main thing for me is that there are two potentially successful versions of this paper: 1. makes a compelling argument for why this method should be used to improve fairness results instead of other existing fairness approaches, and 2. provides insight into this gradient-alignment/fairness hypothesis. I think the current submission could be a basis for either of those papers but at the moment it doesn't currently execute on either: it isn't (1) since it doesn't draw a useful distinction between itself and other fairness approaches, and it isn't (2) since there isn't much exploration (either theoretically or empirically) about why the gradient-alignment/fairness relationship might hold, and what's going on at a deeper level.

**Strengths And Weaknesses:**

The main strength of this paper is the claim that some improvement on fairness metrics can fall out of a MAML formulation/gradient alignment - I think this is interesting and worth exploring. However, I have some doubts about the overall content of the paper and think this exploration needs to be clarified and pushed further to be a truly compelling paper.

My main critiques:
- My main issue with this paper is that it’s not clear why I would use this method over one of the many other ones in the fairness literature that improves these metrics. It’s not clear that there are any distinctive features of this approach that either improves results (either generally, or in some subset of cases) or allows for broader applicability
- On the flip side, I do think that “gradient alignment contributes to fairness metrics improvements” is an interesting conceptual observation, but there is not much work done in this paper to shed light on that insight and so I’m left not really knowing what to take away
- Empirically, the takeaway is a little unclear - for instance, in Table 1, I don’t know if the accuracy hit should be considered “worth it” for the gain in DP. Usually this is resolved by showing a tradeoff between these two metrics along some sort of hyperparameter range, however I’m not sure the MAML approach has such a hyperparameter. All this to say that I think the empirical case could be much more compelling if some argument was made to help me compare the method to the baseline more cleanly
- The SHAP plots seem to suggest that the sensitive attributes (race for COMPAS, gender for Adult) are used - is this true for all experiments? If so, is that required for this method to work? I think it’s non standard for these dataset to include the group labels as model inputs.

Smaller points:
- 2nd paragraph: It’s not clear to me what it means that these methods “treat fairness as an isolated concern rather than a core aspect of the learning process” - I don’t understand the distinction being drawn here
- Intro: distinction btw dataset and feature bias is not clear - it seems like they are different categories of biases (dataset bias can be the root cause of feature bias)
- Sec 4.1, Eq 6 - should clarify what \theta^i is, I don’t think it’s defined explicitly
- Table 4: should clarify why there are so many empty cells
- There’s a claim at the end of 5.5 that increasing the number of iterations improves model fairness. However, in Table 2 this only seems to be true on the COMPAS dataset. For instance, on Adult, if we look at 60:40, we have iter=1 -> Acc=64, DP=0.07, EO=0.07, and iter=10 -> Acc 52, DP 0.1, EO=0.07. This is strictly worse at 10 iterations than at 1

---

> ### Author Response · Authors · 2024-09-11
>
> We thank the reviewer for their detailed and helpful feedback. We appreciate the opportunity to clarify the points raised and have made revisions to enhance the paper's clarity and depth.
>
> 1) **Comparison with Other Fairness Methods:**
> The proposed work differs from traditional approaches since it applies MAML to fairness, a fairly less explored area. We believe this could open doors to further applications of meta-learning algorithms to improve fairness. This method offers a novel perspective by showing that the internal mechanism of MAML can be leveraged to improve fairness metrics from a fundamental level of aligning gradients rather than relying on regularizers that directly target fairness metrics. The proposed method shows big improvements in fairness metrics, which are better than the existing works using meta-learning. It also differs from existing works in the sense that it provides an operating point to vary fairness which could be ideal in real world scenarios.
>
> 2) **Gradient Alignment and Its Impact on Fairness:**
> We have expanded our discussion on how gradient alignment via MAML contributes to fairness improvements in Section 4.1. The latent alignment factor between the query and support sets is responsible for improving fairness across all the experiments. We emphasize how MAML helps in gradient alignment, which has not yet been studied. The updated manuscript includes detailed mathematical proofs and clarifies how aligning gradients between tasks (support and query sets) promotes fairness. Further changes have been made to sec 5.4 to explain how configurations 1 and 2 affect the gradients across the query and support sets.
>
> 3) **Trade-offs:**
> The results in Table 1 are for two specific configurations. In contrast, in Tables 2 and 3, the ablation study has been performed by varying the split of protected samples in the support and query sets. With respect to the proposed method, the ratio of X_s and X_q can be considered as a hyperparameter in the range of 80:20 to 50:50. The ratio could not be taken below the 50:50 ratio because of the restriction in the number of proposed samples in the used datasets. The ablation study in Tables 2 and 3 helps us better understand how adjusting the ratios of samples from different subgroups can serve as a tunable hyperparameter to balance accuracy and fairness. In the updated draft, trade-off graphs have also been added with respect to Table 2 for better understanding.
>
> 4) **Use of sensitive attribute for SHAP plots:**
> The SHAP plots provided were generated during the inference phase and did not influence the training process. We ensured that sensitive attributes were not used as inputs during training to prevent the model from learning discriminatory patterns. The use of sensitive attributes during inference was solely for fairness analysis and understanding the model's decision-making process. This protocol aligns with standard practices in fairness research to diagnose and mitigate bias. Sensitive features were only used during the inference phase across all the experiments of our study.
>
> 5) **Integration of Fairness as a Core Aspect of Learning:**
> In the statement, we intended to highlight that the proposed work integrates fairness during the training process as opposed to many methods that treat fairness as a secondary concern (often addressed post-processing). We aim to ensure that fairness is not just an afterthought but a foundational component of how the model learns from the data. This approach allows us to integrate fairness criteria in the model learning process more deeply.
>
> 6) In response to your suggestion, we have revised Section 4 to define $\theta^i$ and have refined our explanations regarding dataset and feature bias to avoid confusion. The table 4 caption has also been updated to clarify on the number of empty cells in the table.
>
> 7) **Inconsistencies in the result:**
> We acknowledge the observed deviations in the results of Table 2. However, it's important to note that increased iterations leading to improved fairness metrics remain consistent overall (e.g., in Table 1 and most scenarios of Table 2). The deviations observed are partly because of the approximations. The values were rounded, and they might look identical however, they are not.  The results are averaged over the best 5 runs out of 10 independent trails.

---

> > ### Comment · Reviewer_4uSp · 2024-10-21
> > **Thanks**
> >
> > Sorry for the delay here. Thanks for taking the time to address my comments.
> >
> > 1, 2: I appreciate that this a novel application of MAML - however not totally sure that either the advantages or insights are clear from a fairness standpoint yet. However, it's possible that from a meta-learning standpoint this benefit/novelty is more clear.
> >
> > 3: I think tradeoff graphs will be helpful for this type of visualization, thanks.
> >
> > 4. Addressed.
> >
> > 5. I'm not sure I agree with this point - there are many fairness methods that exist which treat fairness as part of the learning process, indeed you cite the category of "in-processing" which I believe refers to this.
> >
> > 6. Thank you, this will help!
> >
> > 7. At a glance, it seems like there are examples throughout Table 2 (e.g. C&C, German) where added iterations make fairness metrics worse. I'm not saying this is a slam dunk argument against the method but I do think that it merits some discussion in the paper if you're going to make the claim that extra iterations are generally helpful.
> >
> > My assessment here is that the proposed amendments will help the paper. From a fairness literature perspective (which I'm more familiar with), I still am unsure that there's enough utility/insight in this paper to merit acceptance; however it's possible that from a meta-learning perspective there is. So I'm leaving my review as is with that caveat.

---

> > > ### Author Response · Authors · 2024-10-24
> > >
> > > We would like to thank the reviewer for their valuable comments.
> > >
> > > 1) Through this work, we intend to demonstrate how the gradient alignment mechanism of MAML, when applied in a fairness context, can provide a more direct approach to improving fairness metrics. In other words, it is a novel approach toward fairness via gradient alignment. We appeal to you to focus more on the fundamentally different way to achieve fairness proposed by the draft. Also, note that the draft fairly supports the following: (i) MAML leading to gradient alignment (section 4.1 of the main draft) and (ii) gradient alignment leading to fairness (multiple sections, e.g., Introduction, 4.1, and configuration-2 in 5.4).
> > >
> > > 2) **In-processing methods:**
> > > We agree with the reviewer that multiple fairness methods treat fairness as a core part of the learning process. Our statement about integrating fairness more deeply emphasizes that we aim to approach fairness through (a novel) gradient alignment within the meta-learning process rather than relying on explicit fairness regularizers. That way, the proposed work falls broadly into the body of in-processing methods, focusing on leveraging meta-learning to address fairness.
> > >
> > > 3) **Trade-off curves:**
> > >  Thank you for the feedback, the trade-off graphs have been included in the updated draft in figure 2.

---

### Review · Reviewer_Edus · 2024-08-23

**Summary Of Contributions:**

Inspired by a meta learning framework (MAML), an inner/outer loop training strategy is proposed in which losses related to protected and unprotected subpopulations are optimized. It is demonstrated that the combined loss can be interpreted as a sum of the loss for the unprotected group and a gradient dissimilarity penalty to the gradient of the protected subgroup loss. Experimental validation on three datasets show that this training strategy leads to lower overall accuracy but strong improvements on two fairness criteria. Several ablations / configurations are included to demonstrate the method. Overall performance is compared with three competing methods.

**Audience:**

Yes

**Claims And Evidence:**

No

**Requested Changes:**

Requested changes

 1. Please provide a clear description of the training algorithm. Is it correctly understood that you alternate between one gradient step on Xq and 1 / 10 gradient steps on Xs?
 2. Please write clearly in the datasets section what is considered the protected subgroup in each case (e.g. in adult, is it men or women?).
 3. Please provide details regarding how the data was split into training and test.
 4. Please include accuracies for all models seperately the protected and the unprotected group (e.g. in table 1).
 5. Please provide details regarding how hyperparameters were chosen, e.g. did you optimize the baseline for accuracy and then reuse the same parameters for the remaining experiments?
 6. Please also provide results for the baseline when doing a manual adjustment (on training or validation data) for demographic parity and equalized odds, to demonstrate whether the proposed method surpasses this baseline in accuracy.
 7. I believe there should be a reference to table 3 in section 5.6.
 8. Please provide comparison with other relevant competing methods on all three datasets. Since all results must be interpreted as a trade-off between accuracy and fairness criteria, I strongly recommend to show curves that demonstrate trade-offs that can be achieved with the different methods. For example, how can we interpret m-FTML vs. FaAlGrad on Adult when the former has better accuracy but worse DPRE and EODR? Showing the entire trade-off curve will provide a much better grounds for comparing the different methods.
 9. "Upon careful evaluation of the results, the proposed method, FaAlGrad, demonstrates notable superiority across all three datasets compared to the aforementioned meta-learning-driven works for improving fairness." I think this is an overstated claim. While results on Communities and Crime is superior, results on Compass and Adult are not evidently better.

Open questions and minor comments

10. It would benefit the paper with a more elaborated discussion of different (conflicting) definitions of bias/fairness.
11. The approach is based on the concept of a protected subpopulation, rather than a protected class, i.e., there is a built-in asymmetry. I would have liked more dicussion of this design choice.
12. Related works section is difficult to read if unfamiliar with the cited work. It is focused on technical differences and does not give much intuition/overview of the conceptual differences between the different approaches.
13. Given that this approach is limited to a single protected subpopulation, can it still be considered meta-learning? It seems to be more similar to a weighted learning scheme?
14. In eq. 6 there is a further restriction to a single gradient update, but in practice you can use more that one gradient update in the inner loop. How does this affect the interpretation of gradient alignment?
15. "This approach Agarwal et al. (2018) framed the challenge of achieving accurate classifiers under fairness constraints as a two-player zero-sum game." Minor grammar issue.
16. "Along similar lines, the work of Slack et al. Slack et al. (2020) delves..." Author name appears twice
17. In the notation starting with Equation 1, would it be clearer to use subscripts like L_T and X_T to specify the loss and data associated with a particular task?
18. The description of MAML could be a bit more concrete. For example, equation 3 suggests the outer loop optimizes an expectation, but figure 1 suggests optimizing only a single inner task.
19. The role of the query data is not discussed in much detail, other than that it "acts as a meta-test". What does that mean in practice?
20. Consider removing line breaks in the derivations below eq. 6 for improved readability.
21. Consider writing explicitly what approximation is made in the second to last line in the derivation on page 6.
22. Is there an approximation made in the last line in the derivation on page 6, or should that be an equals sign?
23. Consider dropping vertical lines from table 1 for improved readability, e.g. use the booktabs latex package.
24. Configuration 2 is not clearly motivated, and only discussed in the results section. Could you provide a more clear argument for this configuration, possibly already in the methods section?
25. Consider just writing "Config. 1" and "Config. 2" in the second column of table 1 in stead of the longer descriptions of Xs and Xq which are clearly described in section 5.4.
26. Consider also showing the results in table 2 as a graph. Consider including the baseline in table 2 as well as in the graph. Consider to include a 90:10 split also.
27. In the experiment reported in table 2, consider leaving out the 10 inner iterations setting, since this experiment is seems to be more focused on reducing "weight" on the protected subpopulation which is countered by a high number of inner iterations.
28. Is there any particular reason for using different fairness metrics in table 4 (DPR and EODR vs. DP Diff and EOD Diff)?

**Strengths And Weaknesses:**

Strengths.
- The paper is generally well written and easy to follow.
- The main idea is simple and seems to work well.
- There is a fair amount of reasonable ablations/experiments included to illustrate the method.

Weaknesses.
- Some details are missing (see below)
- Comparison with other methods is limited. The possible trade-offs between accuracy and fairness is not clearly shown and compared with baselines and other methods.
- Claims regarding the methods superior performance is not sufficiently supported by evidence.

---

> ### Author Response · Authors · 2024-09-11
>
> We thank the reviewer for their valuable time and helpful feedback. We answer the queries in the order of the listed numbers in points and club the points with similar answers.
>
> 1) **Training Algorithm Description**:
> We have elaborated (in the updated draft) on the training algorithm in sec 4. We perform single or multiple gradient steps based on the configuration using samples from the protected class on parameters $\theta_0$ to obtain $\theta^i$ as $\theta^i{=}\theta^{i-1}{-}\eta\nabla_{\theta^{i-1}} \mathcal{L}(\theta^{i-1}, X_s)$. Then, we optimize on the remaining batch samples with the parameters θi and make a final update.
>
> 2) **Protected Subgroup classification**:
> We regret any inconvenience caused by the oversight in the original draft regarding the definition of protected subgroups for each dataset. This detail has now been specified in the datasets.
>
> 3) & 5. **Data split and hyperparameters**:
> The datasets were divided into training, validation, and test sets in a 70%, 15%, and 15% split, respectively. We optimized the baseline models for accuracy and the selected hyperparameters were used for subsequent training of the models. We explored various combinations of inner and outer learning rates to identify the most effective settings. These methodologies are thoroughly detailed in Section 5.3 of the revised draft.
>
> 4) **Accuracies for protected and unprotected group**:
> Due to space constraints, we were limited in how much we could present individual group accuracies in the table. Here are the results on the compass dataset for separate subgroups.
> | No. of inner iterations | X q:X s | Protected accuracy | Unprotected accuracy |
> |---|---|---|---|
> || Baseline| 0.4600 | 0.8113 |
> | 1 | 80:20 | 0.6246 | 0.6706 |
> | 1 | 70:30 | 0.6246 | 0.6246 |
> | 1 | 60:40 | 0.5871 | 0.6554 |
> | 1 | 50:50 | 0.5922 | 0.6416 |
> | 10 | 80:20 | 0.5767 | 0.6326 |
> | 10 | 70:30 | 0.5904 | 0.6427 |
> | 10 | 60:40 | 0.5788 | 0.6495 |
> | 10 | 50:50 | 0.5888 | 0.6405 |
>
> 6) **Baseline comparisons**:
> The results for the baseline models have been added in Table 1. However, it is not possible to include the results for the various splits mentioned in the dataset because the baseline model can not handle multiple sets of data simultaneously, and the union of the query set and support set reflects the original train set itself (refer to sec. 5.5).
>
> 8) &  26. **Trade-off curves**:
> The revised draft now includes trade-off graphs that compare our method with other relevant works and the results presented in Table 2. However, due to a limited number of comparative studies involving the Compass dataset, no specific graph has been included for this dataset in our comparisons.
>
> 9) **Claim of superiority**:
> Our statement emphasizes that our proposed methodology demonstrates superior fairness metrics on the datasets compared to the methods mentioned in the literature. This assertion is focused solely on fairness improvements and does not consider the accuracy metrics.
>
> 11) **Subpopulation vs subgroup**:
> We use the term subpopulation in a generic sense to include a variety of potential subgroupings within the data, which can include multiple attributes of a subgroup.
>
> 13) **Meta-learning**:
>  We clarified that our approach aligns with meta-learning principles, focusing on adaptability across various subpopulations rather than multiple tasks.
>
> 14) **Explanation of gradient updates**:
> In eq 6, a single gradient update was considered to better understand the equations below. However, a new equation has been added in sec 4 to clarify the gradient flow process for multiple updates.
>
> 17) & 18. **Notations in equations**:
> Eq. 1, in general, is for meta-learning. Therefore, the standard notations of MAML were used. The figure, however, specifically illustrates our proposed method, which focuses on a singular inner task tailored for the query set that exclusively contains protected samples. This is why there is a slight mismatch of notations in the MAML section and the figure.
>
> 19) **Role of query set**:
> The role of the query set in the context of meta learning has been mentioned in sec 3.1. However, for the proposed work, the role of the query set is to generalize better on the entire dataset. In terms of gradient alignment, the dot product in gradients of query set and support set is important (sec 4.1) because it ensures that the bias-reducing updates from the inner loop remain effective when the model is tested on a query set that would typically reinforce bias.
>
> 7) , 15., 16. & 20-25. **Minor corrections and updates**:
> In response to your suggestions, we have revised the draft to include the required information.
>
> 28) **Change in fairness metrics**:
> We have adjusted the fairness metrics to align with those used by the studies we are comparing against to ensure that our results are directly comparable. Although, we believe that they essentially convey the same thing.

---

> > ### Comment · Reviewer_Edus · 2024-09-17
> >
> > Thank you for taking the time to address my comments. I appreciate the improvements made, though there are still a few points that I believe could use further clarification (numbered according to my original requested changes):
> >
> > 1. Addressed
> > 2. Addressed
> > 3. I believe there are still some details missing, for this to be reproducible. If the splits are random, do you provide code to reproduce the splits?
> > 4. Addressed, but this should be included in the paper, since without this information it is not possible to assess the tradeoff between accuracy in protected/unproteced classes versus the overall fairness criterion.
> > 5. It is still not clear to me from the description in the paper how the hyper parameteres were optimized for each setting.
> > 6. I believe this point was not addressed?
> > 7. Addressed
> > 8. The curves in figure 2 and 3 are resonable, but do not provide much information beyond what is already in the results table. My point was that in order to be able to compare these methods, which all can attain different trade-offs between accuracy and fairness, you should plot entire trade-off curves, to see if any method dominates. With just a single point per method, there is no basis for comparison.
> > 9. I still think this is an over-stated claim. Depending on the fairnes metric in question, we can achieve arbitrarily good fairness if we are willing to sacrifice accuracy.

---

> > > ### Author Response · Authors · 2024-10-11
> > >
> > > We thank the reviewer for their valuable time and insightful comments/suggestions.
> > >
> > > 1) **Code to reproduce splits and results:**\
> > > A link to the anonymized GitHub repository has been included in the updated draft, which provides the code necessary to reproduce the results for various splits mentioned in Table 1 and 2.
> > >
> > > 2) **Accuracy of protected/unprotected group:**\
> > > Following the suggestion to include the accuracy results on different subpopulations in the paper, we have added Table 3 in the paper, which presents the accuracies across protected and unprotected groups. The results from the table suggests the significant improvement in accuracy across the protected group as compared to the baseline.
> > >
> > > 3) **Hyperparameter-optimization:**\
> > > We optimized the baseline models for accuracy, and the selected hyperparameters were then used for subsequent training. We explored various combinations of inner and outer learning rates to identify the most effective settings. These hyperparameters were optimized using cross-validation. While the remaining hyperparameters such as network architecture and activation functions used were kept consistent with those used in the baseline models to maintain a fair comparison. This has also been updated in section 5.3 in the draft.
> > >
> > > 4) **Manual adjustment of baseline:**\
> > > We would like to thank the reviewer for suggesting the use of manually adjusted baselines for comparison. We have now included results for the manual adjustment of the baseline on training data for both demographic parity and equalized odds. DP baseline used ensures equal representation of samples from different subpopulations in the training data whereas the EOD baseline ensures equal True Positive Rate (TPR) and False Positive Rate (FPR) across protected and unprotected groups. While, in some cases the manually adjusted baseline improves the results, the proposed work clearly achieves superior results.
> > >
> > >
> > > 5) **Trade-off comparison with existing works:**\
> > > Thanks for clarifying, we misunderstood the comment by the reviewer earlier. The existing works cited in the comparison section use meta learning but do not specifically focus on optimizing fairness across a range of operating accuracies. In other words, the proposed method provides different operating tuples of accuracy and fairness, which may not be the objective of the existing meta learning driven works. In summary, these works do not provide a trade-off for accuracy and fairness. Hence, plotting trade-off curves was not feasible. However, Table 2 in the draft demonstrates the trade-offs between accuracy and fairness.

---

> > > > ### Comment · Reviewer_Edus · 2024-10-22
> > > >
> > > > 1. **Code to reproduce splits and results:** Thank you, this makes the results reproducible.
> > > > 1. **Accuracy of protected/unprotected group:** Thank you - these results certainly provide more insights. However, I am not sure how the baseline accuracies could be so skewed? I would expect accuracies to be very similar to for the two groups (absent any fairness adjustment) since the baseline is only optimized for accuracy? I ran a logistic regression on the compass dataset which gives an overall accuracy of 0.6737 and (0.6644 vs 0.6823) on the protected vs unprotected groups which is in line with my assumption.
> > > > 1. **Hyperparameter-optimization:** Thank you. This is more clear now.
> > > > 1. **Manual adjustment of baseline:** Thank you. Your approach makes sense to me, but it was not exactly what I meant. Let me clarify with an example: DP requires equal distribution of the positive predictions, and let us say that the protected class has "too few" positive predictions.  DP can then be achieved by predicting more positives for the protected class or less positives for the unprotected class or somewhere in between. This can manually be adjusted by modifying the thresholds for the classifier. But of course this comes at the cost of reduced accuracy. For example, with a logistic regression model, adjusting just the threshold for the protected class, we can achieve an overall accuracy of 0.6397 and (0.5940 vs 0.6823) on the protected vs unprotected groups. This approach trivially achieves a zero DP diff. Incidentally it also achieves a fairly good EO diff of 0.05. I would expect to see not only such types of baselines, but also plots of trade-off curves that show which accuracies and privacy metrics are attainable.
> > > > 1. **Trade-off comparison with existing works:** I disagree. I think this issue is central. Without trade-off curves there is no clear grounds for comparison. At least you could provide detailed trade-off curves for your proposed approach as well as simple baselines as the one I suggested above.

---

> > > > > ### Author Response · Authors · 2024-10-24
> > > > >
> > > > > We thank the reviewer for their valuable feedback and time spent reassessing our feedback.
> > > > > 1) **Accuracy of protected/unprotected group:**
> > > > >  Thank you for pointing out the discrepancy in the baseline accuracies. Upon review, we identified an entry error that resulted in skewed accuracy values for the protected and unprotected groups. We have corrected this in the updated draft, and the accuracy for the protected and unprotected groups is now more balanced, as expected from a baseline optimized for accuracy.
> > > > >
> > > > > 2) **Manual adjustment of baseline and trade-off curves:**
> > > > >  We appreciate your clarification on the manual threshold adjustment baseline. We implemented a threshold-based baseline where classification thresholds (on the activation of the output sigmoid neuron of the neural network) ranging from 0.1 to 1.0, with a step size of 0.1, were applied for the protected group. This allowed us to fine-tune the predictions and achieve better fairness in terms of DP diff. As expected, the threshold adjustments improve DP diff significantly; however, this comes at the cost of accuracy, and the Equalized Odds (EO) difference remains poor due to the threshold adjustment focus on DP. In Table 1 of the updated draft, we provide the accuracy and DP difference values for the threshold that minimizes DP diff, as including metrics for all thresholds was not feasible. Additionally, trade-off curves illustrating the DP diff versus accuracy across different thresholds are now presented in the updated draft (Figure-2) , clearly showing the impact of threshold adjustments.

---

### Review · Reviewer_Er9X · 2024-09-27

**Summary Of Contributions:**

This paper introduces a new approach to mitigating bias in machine learning models by using a gradient-alignment strategy based on the Model-Agnostic Meta-Learning (MAML) framework. By aligning loss gradients computed across protected and unprotected populations, the paper proposes a method to regularize the training process for fairness without using any specific fairness regularizers. The experiments are conducted on multiple commonly used benchmark datasets (COMPAS, Adult, and Communities and Crime) and they show improvements in fairness metrics like demographic parity and equalized odds, with the usual trade-offs in accuracy. The core contribution of the work is the integration of fairness considerations directly into the learning process (in/meta-process), as opposed to pre/post-processing techniques.

**Audience:**

Yes

**Broader Impact Concerns:**

COMPAS dataset  has been discussed several times to be, at least, controversial for benchmarking usage for Responsible AI methods. See “It's COMPASlicated: The Messy Relationship between RAI Datasets and Algorithmic Fairness Benchmarks” (https://arxiv.org/abs/2106.05498), or “The Age of Secrecy and Unfairness in Recidivism Prediction“ (https://hdsr.mitpress.mit.edu/pub/7z10o269/release/7)

**Claims And Evidence:**

Yes

**Requested Changes:**

- Adding a paragraph discussing why COMPAS dataset is used, and warning about the potential issues of it (see Broader impact concerns)
- Considering COMPAS issues, adding another dataset might beneficial for the paper, such as the “German Credit Dataset”

**Strengths And Weaknesses:**

Strengths:
- The paper tackles the problem of Responsible A.I. for ML models, which is an important issue
- The idea of aligning gradients across subpopulations is sufficiently innovative
- The results prove the benefits in fairness of the proposed approach compared to other baselines, with a reasonable trade-off in accuracy

Weaknesses:
- The number of dataset for Responsible AI benchmarking is limited, and this is a well known issue. I suggest to add at least all the common ones, adding “German Credit Dataset”. Moreover, see the section “Broader impact concerns” for further details on datasets.

Minors/Typos:
- Use one between subpopulations or sub-populations, not a mix of the two. I suggest “subpopulations”.
- Page 9: in hand -> at hand
- Page 11: Differnce -> Difference

---

> ### Author Response · Authors · 2024-10-11
>
> We thank the reviewer for their valuable time and insightful comments/suggestions.
>
> 1) A caveat and detailed description regarding using the Compas dataset has been added to the draft in Section 5.1. This update acknowledges the known issues and broader impact concerns related to the dataset.
>
> 2) Following the suggestion, the German Credit Dataset has been incorporated into the analysis, and the results have been added to the draft in Table 1 and 2. The results demonstrate that the proposed method achieves significant improvement in fairness without compromising on the predictive accuracy.
>
> 3) All other minor revisions suggested have also been implemented.

---

> > ### Comment · Reviewer_Er9X · 2024-10-17
> >
> > Thank you for taking the time to address my comments.
> >
> > Minor: In Table 1 -- in the new section for the "German Credit Dataset" -- there is two times a row called "MBA for DP", I suppose the second is indeed "MBA for EOD".

---

> > > ### Author Response · Authors · 2024-10-21
> > >
> > > Thank you for your feedback and attention to detail. We apologize for the oversight and the correction has been implemented in the updated draft.

---

### Decision · Action_Editor_9DBC · 2024-12-10

**Recommendation:** Accept as is

**Comment:**

The authors addressed all the comments raised by the reviewers. In particular, they conducted additional experiments as requested and included additional material to clarify the open questions. Importantly, they also provided trade-off curves as requested by two out of the three reviewers. This was one of the key concerns. The authors also provided code and documentation to reproduce the results.

**Audience:**

Sharing methods and encouraging experimental results that borrow ideas from the meta-learning literature to improve fairness in ML is of interest to the community. As acknowledged by the reviewers, this is valuable and other work in this space is sparse.

**Claims And Evidence:**

The contributions of this paper are modest. Given that novelty is not central to TMLR acceptance criteria, I recommend acceptance as the claims in the revised version of the paper are supported. The core idea of the paper inspired by MAML is clearly detailed and experimental results validate it on a number of public benchmarks. The authors clarified the questions and concerns raised by the reviewers in a satisfactory manner. While insights are modest, the conclusions show that the core idea can be beneficial in the experiments conducted.